# Relationship between Mediterranean Diet Adherence and Saliva Composition

**DOI:** 10.3390/nu13041246

**Published:** 2021-04-10

**Authors:** Teresa Louro, Carla Simões, Maria João Penetra, Laura Carreira, Paula Midori Castelo, Henrique Luis, Pedro Moreira, Elsa Lamy

**Affiliations:** 1MED (Mediterranean Institute for Agriculture, Environment and Development), Institute for Advanced Studies and Research (IIFA) IIFA—Instituto de Investigação e Formação Avançada, University of Évora, 7002-556 Évora, Portugal; teresalouro@hotmail.com (T.L.); carlasimoes3@hotmail.com (C.S.); d47128@alunos.uevora.pt (L.C.); 2Department of Biology, University of Évora, 7002-556 Évora, Portugal; l37950@alunos.uevora.pt; 3Department of Pharmaceutical Sciences, Universidade Federal de São Paulo (UNIFESP), Diadema 09972-270, Brazil; pcastelo@yahoo.com; 4Unidade de Investigação em Ciências Orais e Biomédicas (UICOB), Faculdade de Medicina Dentária, Universidade de Lisboa, 1600-277 Lisboa, Portugal; henrique.luis@fmd.ulisboa.pt; 5Center for Innovative Care and Health Technology (ciThecCare), Politécnico de Leiria, 2411-901 Leiria, Portugal; 6Faculdade de Ciências da Nutrição e Alimentação da Universidade do Porto, 4150-180 Porto, Portugal; pedromoreira@fcna.up.pt

**Keywords:** dietary polyphenols, Mediterranean diet, salivary amylase, salivary cystatins

## Abstract

Dietary polyphenol exposure is known to change protein saliva composition in rodents, but less is known in humans. The present study aimed to assess the relationship between saliva protein composition and adherence to Mediterranean Diet (MD) and polyphenol intake levels. Participants were assessed for their dietary habits, which were converted in Mediterranean adherence level, according to Mediterranean Diet Adherence Score (MEDAS) score. Total polyphenol and total flavanol intakes were extrapolated from dietary data, using Phenol explorer database. Whole saliva was collected, and proteins were separated by SDS-PAGE. Salivary S-type cystatins were highly expressed in the group with medium adherence to MD, being positively correlated with wine intake in overweight individuals. The association between salivary amylase and MD adherence also depended on Body Mass Index (BMI), with a positive association only in normal weight individuals. Polyphenol intake was positively associated with S-type cystatins levels, particularly when flavanols were considered separately. These results show that saliva relationship with MD adherence depend on BMI, suggesting that normal weight and overweight individuals may have different salivary responses to diet. Moreover, these results reinforce the link between saliva and dietary polyphenols (flavanols) levels, leading to the hypothesis that salivary proteome can have a role in polyphenol-rich foods acceptance.

## 1. Introduction

Mediterranean Diet (MD) is accepted as one of the most healthy and sustainable dietary patterns. High adherence to MD has been shown to relate with several different health benefits, such as obesity reduction, diabetes, and cardiovascular protection, among many others [1,2]. Despite of these advantages, only a minimal percentage of the population has high adherence to MD, even in Mediterranean countries [3,4,5]. In a recent study made for the Portuguese population, participants reported different reasons to justify the medium-low adherence levels, among which is food’s flavor [6].

The MD dietary pattern is characterized by the intake of considerable amounts of vegetables, fresh fruits, and whole grain cereals, with moderate to low amounts of products of animal origin, having extra-virgin olive oil as the main source of fat. In addition, low to moderate consumption of wine, at main meals, is also present in MD dietary pattern. Altogether, this makes MD a polyphenol rich diet [7]. Polyphenols are constituted by different groups of compounds. Flavanols, or flavan-3-ols, are a group of polyphenols that may be present in vegetable foods in monomeric (catechins) and polymeric (proanthocyanidins) forms. This group, which is abundant in foods, like cocoa, tea, grapes, and wine, is responsible for the bitterness and/or astringency of these foods. As such, despite the health advantages, it may limit sensory quality.

Saliva is a biological fluid that plays numerous roles in the oral cavity, participating in oral food perception [8]. The interaction between salivary proteins and dietary polyphenols has been studied for several years and it is well known to be responsible for the astringency perceived for these dietary compounds and this, consequently, influences their acceptance [9,10]. Furthermore, salivary protein composition is also related with bitterness perception [11], which can also affect the acceptance of polyphenol-rich foods. Recent studies, performed in animal models, described an association between the levels of salivary cystatins and the acceptance of bitterness and polyphenol rich foods [12], supporting this hypothesis.

Besides the potential effect of saliva on acceptance and intake of polyphenol rich foods, it should also be simultaneously considered the potential effect of polyphenols ingestion in salivary protein composition. In different animal studies, it was observed that increasing the levels of dietary tannins, a particular type of polyphenols, results in changes in salivary proteome [13,14]. Although the effect of polyphenol ingestion in human salivary protein composition has been less deeply investigated, the results of some studies suggest this possibility. Short term effects were observed immediately after ingestion of a tannin solution [9] or wine (own data, not published). In addition, a recent study, where chocolate milk with polyphenols was tested, suggested an effect of these compounds in saliva proteome [15].

For all that was stated above, namely the potential influence of saliva protein composition in the acceptance of polyphenol rich diets, on one hand, and the possible effect of polyphenol intake in saliva protein composition, we hypothesize that the adherence to a Mediterranean dietary pattern is associated with polyphenol intake and with saliva composition. This study was performed with the general objective of assessing how MD adherence is related with saliva protein composition. Since MD food products are rich in polyphenols, two specific objectives were considered: (1) to assess the differences in salivary protein profile among groups with different adherence levels to MD; and (2) to investigate if salivary protein profiles relate with the levels of total polyphenols and flavanols in diet.

## 2. Materials and Methods

### 2.1. Participants and Anthropometric Parameters

One hundred twenty-two adult individuals (61 from each sex, with 18 to 65 years old) participated in this study. Only individuals not reporting oral or nasal health problems, nor taking medication, were considered. Participants were instructed to arrive at test room between 10:00 a.m. and 11:00 a.m., at least 1 h and 30 min after breakfast intake, and without eating or drinking anything other than water, after that.

Height and weight were measured according to the European Health Examination Survey procedures, with participants evaluated in a stand position wearing light cloths and barefoot. Height and weight were accessed using a portable stadiometer (Seca 214) and a digital scale accurate to the nearest 0.1 kg (Seca 803), respectively. Body mass index (BMI) (kg/m^2^) was used to classify individuals in underweight, normal weight, pre-obese, and obese, accordingly: BMI < 18.5; 18.5 < BMI < 24.99; 25 < BMI < 29.99; and BMI > 30 [16].

A description of the experiment and an informed consent form were provided to all participants, before the beginning of the study, being read and signed by all of them. All procedures were performed according to the Declaration of Helsinki for Medical Research Involving Human Subjects and had ethical approval from the Ethical committee of the University of Evora.

### 2.2. Mediterranean Diet Adhesion Patterns Assessment

The participants data, analyzed in this study, is part of a larger study, developed between 2017–2018, aimed to assess the relationship between dietary patterns and oral food perception (data analysis in course). Frequency of food consumption, over the previous 12 months, was assessed using a self-administered, semi-quantitative food frequency questionnaire (FFQ), validated for Portuguese adults [17], which included 86-food items. Before questionnaire completion, participants were instructed on how to fill it. Food intake was calculated by multiplying one of the nine possibilities of frequency of consumption (from—never or less than once per month, to six or more times a day), by the weight of the standard portion size of the food-item. For energy and nutritional intake estimation, an adapted Portuguese version of the nutritional analysis software Food Processor Plus (ESHA Research Inc., Salem, OR, USA) was used. Mediterranean Diet adherence was obtained through the parameters considered in Mediterranean Diet Adherence Score (MEDAS) questionnaire, which was created by PREDIMED (Prevención con Dieta Mediterránea) study [18]. The MEDAS is a 14-item questionnaire where participants are requested to report their habitual frequency of consumption or amount consumed of 12 foods characteristics of MD and two cooking habits related to MD. Each of these items is scored 1 or 0, depending on their positive relation to a MD pattern. The scores were obtained calculating them from the FFQ, using the criteria previously used in the Portuguese population [19]. For categorization of the adherence to MD (MD adherence groups), we used the criteria previously described [18], according to the total Med scores (points) obtained: weak adherence, ≤5; moderate to fair adherence, 6–9; good or very good adherence ≥10.

Since the FFQ does not access polyphenols consumption, an extra questionnaire, developed using the same model of the validated FFQ was applied, in which 50 fruits and vegetables were presented individually. The amount (per day) of each food item was calculated by multiplying the frequency of consumption (number of days per month/week) by the weight of the standard portion size of the food-item, as previously described for FFQ. The amounts of polyphenol and flavanol of each food considered (each individual fruit and vegetable, olive oil, wine, chocolate, and coffee/tea) were extracted from a polyphenol database [20]. Daily intakes of these compounds were obtained by multiplying the daily consumption of each food by its amount per weight of total polyphenols or flavanols.

### 2.3. Saliva Collection and Cleaning

Unstimulated saliva was collected, from each participant, by accumulating all saliva produced in the mouth and spitting it to a tube, maintained on ice, at regular intervals, for 3 min. Before saliva collection, individuals were asked to rinse their mouth with water to remove residual saliva and/or any food debris. After collection, saliva was transported, on ice, to laboratory, where it was frozen at −28 °C. With the aim of removing mucins, cells and other solid material, saliva samples were thawed on ice and centrifuged at 13,000 *g* for 30 min at 4 °C. The supernatant was recovered and stored at −80 °C until subsequent analysis.

### 2.4. Saliva Laboratory Analysis

#### 2.4.1. Saliva Flow Rate, pH, and Total Protein Concentration

Saliva flow rate was calculated by assuming a density of 1.0 g/mL for saliva. Tubes containing saliva were weighed and the weight of the empty tube was subtracted. In order to calculate flow rate, the final volume was divided by the number of minutes of saliva collection. Saliva pH was measured using a calibrated pH meter (Hanna Instruments, Póvoa de Varzim, PT) and recording to two decimal places. The Bradford method was used for total protein concentration determination. Bovine serum albumin (BSA) was used as standard, and the method was carried out in microplates of 96-weels, which were read at 600 nm in a microplate reader (Glomax, Promega, Madison, WI, USA).

#### 2.4.2. SDS-PAGE Separation and Protein Profile Analysis

Each saliva sample was run in 14% polyacrylamide gels in duplicate. The volume of each sample used corresponded to 7.5 µg total protein. Samples were mixed with sample buffer and run on each lane of a mini-gel (Protean xi, Bio-Rad, CA, USA) using a Laemmli buffer system, as described before [21]. Proteins were fixed by having gels in 40% methanol/10% acetic acid, for 1 h, after which gels were changed to a solution of Coomassie Brilliant Blue (CBB) G-250 (2% in 40% methanol, 10% acetic acid), for 2 h, followed by destaining in different water changes. Gels were scanned for image acquisition using a scanning Molecular Dynamics densitometer with internal calibration and LabScan software (GE Healthcare Europe GmbH, Freiburg, Germany). Gel images were analyzed using GelAnalyzer software (GelAnalyzer 2010a by Istvan Lazar, www.gelanalyzer.com, accessed on 1 May 2019) and values of volume for each band were normalized for analysis of the volume percentage of each protein band. Molecular masses were determined in accordance with molecular mass standards (Bio-Rad Precision Plus Protein Dual Color 161–0394) run with protein samples.

#### 2.4.3. Salivary Amylase Enzymatic Activity

A Salimetrics^®^ kit was used to determine the enzymatic activity of salivary amylase according to the manufacturer’s recommendations and following the procedure described before [22].

### 2.5. Statistical Analysis

Exploratory analysis consisted of means, standard deviation, medians and quartiles. Age, BMI and the total amount of polyphenol intake were compared among MD adherence groups using One-way ANOVA. The relationship between the adherence to MD diet and total polyphenol consumption was accessed using the Spearman correlation test between total Med scores and total polyphenol daily intake. The differences between sexes in MD adherence was compared using the Chi-squared test.

Salivary bands A-H, I1, I2, J, and K, and amylase/protein (U/mg) concentration were compared between MD groups and BMI groups using the Two-way ANOVA test, as well the interaction effect MD groups*BMI groups. The results of Levene’s equality of variance and normality tests were evaluated as ANOVA assumptions. A Bonferroni-type adjustment was made to prevent alpha inflation; the effect size and power of the test were also examined for interpretability purposes.

Principal component analysis was used to estimate the number of latent variables emerging from the salivary SDS-PAGE bands (A-H, I1, I2, J, and K), amylase/protein (U/mg), and flavonoids and polyphenol intake, to derive optimal non-correlated components among the parameters. As the variables showed moderate correlations, the Oblimin rotation was performed. The overall Kaiser-Meyer-Olkin (KMO) measure and Bartlett’s test of sphericity were examined which are required for a good principal component analysis, and the decision of the number of components to be retained was based on eigenvalues, total of explained variance and Scree plot examination. Statistical analysis was performed using SPSS 27.0 software.

## 3. Results

### 3.1. MD Adherence Participants’ Characteristics

The higher percentage of individuals participating in this study presented medium adherence to MD pattern (62.3%), with 14.8% of the individuals having poor and only 22.9% high adherence, respectively.

When comparing the groups of different adherence levels, for their total polyphenol intake, it was observed increasing amounts consumed as the adherence to this dietary pattern increased, with mean values of 0.88 ± 0.43, 1.5 ± 0.9, and 2.19 ± 0.88 mg epicatechin equivalents/100 g food for low, medium and high adhesion patterns, respectively. This increase in the levels of total polyphenol intake, as MD adherence levels increase, was corroborated by a moderate statistically significant correlation, observed among these two parameters (rho = 0.487; *p* = 0.0005) (Figure 1).

When the three groups of MD adherence were compared for age or BMI, no differences were observed. Concerning sex, a higher adherence was observed in women (8.8%, 56.1%, and 35.1% of women having low-, medium-, and high adherence, respectively), comparatively to men (20.0% 68.3%, and 11.7% of men having low-, medium-, and high-adherence, respectively) (*p* = 0.006; Chi-squared test).

### 3.2. Variations in Saliva Composition among MD Adherence Groups

Electrophoretic separation of salivary proteins resulted in the observation of 12 protein bands, in the molecular mass range of 14–110 kDa, consistently present in the different individuals (Figure 2), being that these were the ones subjected to analysis.

Table 1 shows the effect of the factors MD adherence groups and BMI groups, and the interaction MD adherence groups*BMI groups, on the expression levels of salivary bands A-H, I1, I2, J, and K, and on the enzymatic activity of amylase/total protein (U/mg).

According to the results of the Two-way ANOVA test, a significant effect of MD adherence groups was observed on band K (previously identified to contain cystatins [23]): Group 2 (medium adherence) was different from group 1 (low adherence) and group 3 (high adherence) (*p* = 0.005; Eta partial squared = 0.09; power of the test = 85%), with a medium effect size (Figure 3). Additionally, a significant interaction MD adherence groups*BMI groups effect was observed for amylase enzymatic activity/total protein, what means that a difference was observed considering the adherence to MD depending on BMI. Figure 4 illustrates these results showing that the MD adherence groups 1 and 3 differed between them, only in normal-weight participants (*p* = 0.019; Eta partial squared = 0.11; power of the test = 80%), with a medium effect size.

To better understand the reason for higher levels of band K in the group with medium MD adherence, an association between the levels of this band and the consumption of foods, considered in each of the 14 items of MEDAS scale, was assessed. Correlations were observed for non-olive oil fats and wine, but only considering overweight individuals (rho = −0.250; *p* = 0.036; *N* = 70 and rho = 0.231; *p* = 0.053; *N* = 70 for non-olive oil fat and wine, respectively). Considering only obese individuals the strength of the correlation increases and in the case of wine, it becomes statistically significant (rho = −0.415; *p* = 0.025; *N* = 29 and rho = 0.381; *p* = 0.042; *N* = 29, for non-olive oil fat and wine, respectively).

### 3.3. Association of Saliva Composition and Polyphenol/Flavanol Intake

A PCA was run to extract latent components emerging from the salivary bands A-H, I1, I2, J, and K, amylase/protein (U/µg), total polyphenol, and flavonoids intake. The suitability was assessed prior to analysis, and the inspection of the correlation matrix indicated those variables that had at least one correlation coefficient greater than 0.30. The KMO measure was 0.56, and the Bartlett’s test of sphericity was statistically significant (χ^2^(105) = 273.4; *p* < 0.0001). PCA retained five components that had eigenvalues greater than one and which explained 57% of the total variance, as confirmed by visual inspection of the scree plot (Figure 5). As such, five components met the interpretability criterion.

As shown in Table 2, the interpretation of the data gathered after Oblimin rotation was consistent with the attributes of the involved aspects: the higher the Component 1, the lower band B, G, and K and higher band C, D, F, and enzymatic activity of amylase/total protein (U/µg); the higher the Component 2, the higher flavonoids and polyphenol intake and lower band F; the higher the Component 3, the higher band A, C, H and lower band K; the higher the Component 4, the higher band C, F, J, K, and flavonoids intake; finally, the higher the Component 5, the higher band E, G, I1, I2, and lower band H and K.

## 4. Discussion

Saliva composition has been greatly recognized to be linked to diet. On one hand, it can influence oral food perception [11] and, consequently, food choices; on the other hand, it can be influenced by the types of foods frequently eaten [24]. Since changes in salivary protein profile, induced by the presence of polyphenols in diet, have been reported for different animals (e.g., References [12,14]), and since human salivary proteins react with polyphenols [25,26], it is possible to hypothesize that individuals with different levels of polyphenols, in their regular diets, present differences in their salivary protein composition. For this reason, the present study compared individuals with different levels of adherence to MD.

From the individuals participating in this study, only 22.9% presented high adherence to MD, with the majority of participants presenting medium adherence levels to this dietary pattern. The daily intake of polyphenols was significantly higher in the individuals with high adherence to MD. This was somehow expected, since most of the main representative foods of this dietary pattern are rich in polyphenols, such as olive oil and olives, vegetables, fruits and pulses, and wine [18]. In the present study no significant differences in mean BMI were observed between high and low MD adherence groups, despite MD pattern had been linked to lower levels of obesity in different studies (e.g., [18,27,28]). These different results may be due to the limited number of individuals with extreme adherence groups (17 for low and 28 for high), that we had in this study. Despite these lack of differences in BMI among MD adherence groups, BMI was negatively associated with total polyphenol intake, in line with the studies reporting the positive effects of those plant secondary metabolites in obesity and metabolic diseases [29].

The protein band K presented higher expression levels in individuals with intermediate adherence to MD, being similar for low and high MD adherence groups. This protein band was previously identified as containing salivary cystatins. Why this happens is not clear, but it was interesting to observe that this saliva association with MD only occurs in overweight (pre-obese and obese) individuals. Cystatins are cysteine protease inhibitors, for which an association with diet and oral food perception has already been observed in different studies (e.g., References [11,12,30,31]). In order to understand the reason why these proteins could be increased only in the overweight individuals with an intermediary MD adherence level, each of the 14-itens used to achieve MEDAS-score was searched for a potential relationship with band K, emerging the evidence that the expression levels of this band were negatively correlated with non-olive oil fat and positively correlated with wine consumption. Wine is a product rich in tannins, a particular type of polyphenols recognized by their high affinity to complex and bind salivary proteins [25,26] and increased levels of dietary tannins result in changes in saliva protein composition, at various levels [14,32], including increases in salivary cystatins [12,33]. The type of stimulation that result in the salivary protein changes in consequence of tannin ingestion is not well known, but it is clear that these are astringent compounds. Moreover, flavanols were also observed to be positively associated with these salivary proteins, in the present study, as further discussed, and proanthocyanidins are present in red wines. Concerning non-olive oil consumption, although no studies were found that report association between it and salivary cystatins, the liking for fat was reported to be positively associated with proteolysis activity of saliva. Since cystatins are protease inhibitors, this negative association between cystatins and non-olive oil fat intake may mean a positive association between the saliva proteolytic activity of these individuals and non-olive oil fat intake. Another question that emerges is “why cystatins are only related with non-olive oil fat intake, rather than with total fat intake?”. In this case, the fact of olive oil having the particularity of being polyphenol-rich may be the explanation. But, further studies are needed here to validate these hypotheses.

PCA analysis allowed to observe association between salivary cystatins and polyphenols. But, at the same time, this analysis allowed to elucidate that the association was not with the total of polyphenols, but rather with the total of flavanols. As referred before, flavanols constitute a class of polyphenols that includes catechins and pro(antho)cyanidins, which are responsible for bitterness and astringency [34], reinforcing the link between salivary cystatins and bitterness/astringency [33,35].

The relationship between MD adherence and saliva was observed also at the level of salivary amylase. As in the case of cystatins, this relationship was also influenced by BMI, with increasing levels of the protein observed only in normal weight individuals. It could be hypothesized that such increase with higher levels of MD adherence could be related with the increased levels of polyphenols associated, since salivary amylase was one of the proteins induced by polyphenol (tannin) intake [14,32]. However, it was not observed a significant relationship between amylase and total polyphenols, neither between amylase and total flavanols, suggesting that other factors, besides polyphenols, may influence the link between amylase and MD adherence. A link between salivary amylase and starchy-rich foods intake was previously reported [36]. Cereals, including the starchy-rich ones are also part of MD, but they were not considered in any of the 14-points of MEDAS-score. So, a higher intake of starchy-rich foods for the individuals with high MD adherence cannot be excluded.

The effect of BMI in salivary amylase is particularly interesting. Some controversy exists if overweight/obese and normal weight individuals differ in salivary amylase amounts. Whereas some studies observed higher levels of amylase in the whole saliva of obese [21,37], others report lower levels [38,39] and others no significant differences [40]. But, despite these inconsistent results for salivary levels, some studies found lower copy numbers of the gene that codifies for salivary amylase in obese individuals [41]. The exact relationship between the number of copies of the gene and the protein expression is not clear, but it is known that higher numbers of gene copies mean higher capacity of stronger transduction and consequent protein synthesis. As such, it is possible that diet have lower impact in salivary amylase levels in overweight/obese individuals, due to a lower response by these ones. This needs to be explored in further studies.

## 5. Conclusions

From our knowledge, this is the first time that saliva is studied in the context of MD diet. The results of this study show association between saliva protein composition and this dietary food pattern, but this appears to differ in function of the food items contributing to that adherence level, showing that different salivary proteins can better indicate the intake level of particular foods or food groups, rather than indicate the total level of adherence.

It seems to be the flavanol group that mostly contribute to the relationship between saliva and polyphenol intake. Salivary cystatins appear to have potential as biomarkers of these bitter/astringent compounds intake. Nevertheless, it is important to highlight that the unbalanced number of individuals belonging to each MD adherence group, due to different distribution by each group, is a limitation, due to the reduced number of participants from the high and low adherence groups.

One interesting observation, in this study, is the influence that BMI has in the relationship between saliva and diet. This is particularly relevant, not only because it shows that overweight individuals can respond to diet differently than normal weight ones but also because it reinforces the need of considering BMI when studying saliva.

## Figures and Tables

**Figure 1 nutrients-13-01246-f001:**
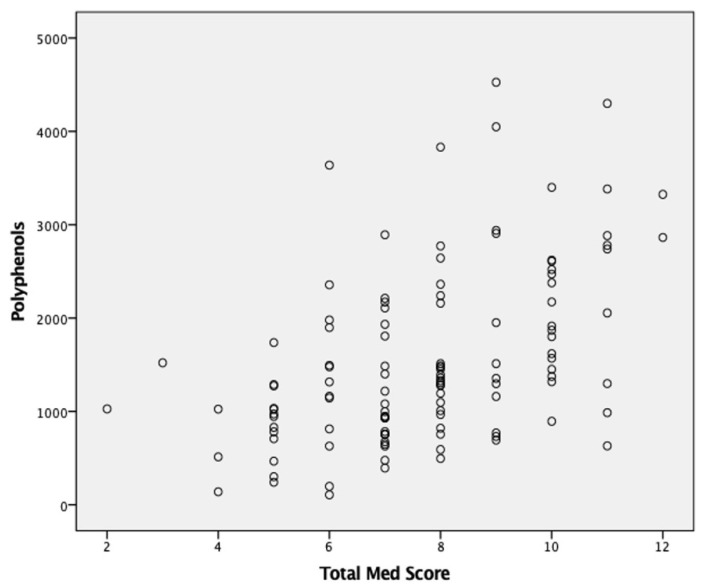
Scatter plot between total polyphenol intake (mg epicatechin equivalents/100 g food) and total Med Score [obtained from Mediterranean Diet Adherence Score (MEDAS)].

**Figure 2 nutrients-13-01246-f002:**
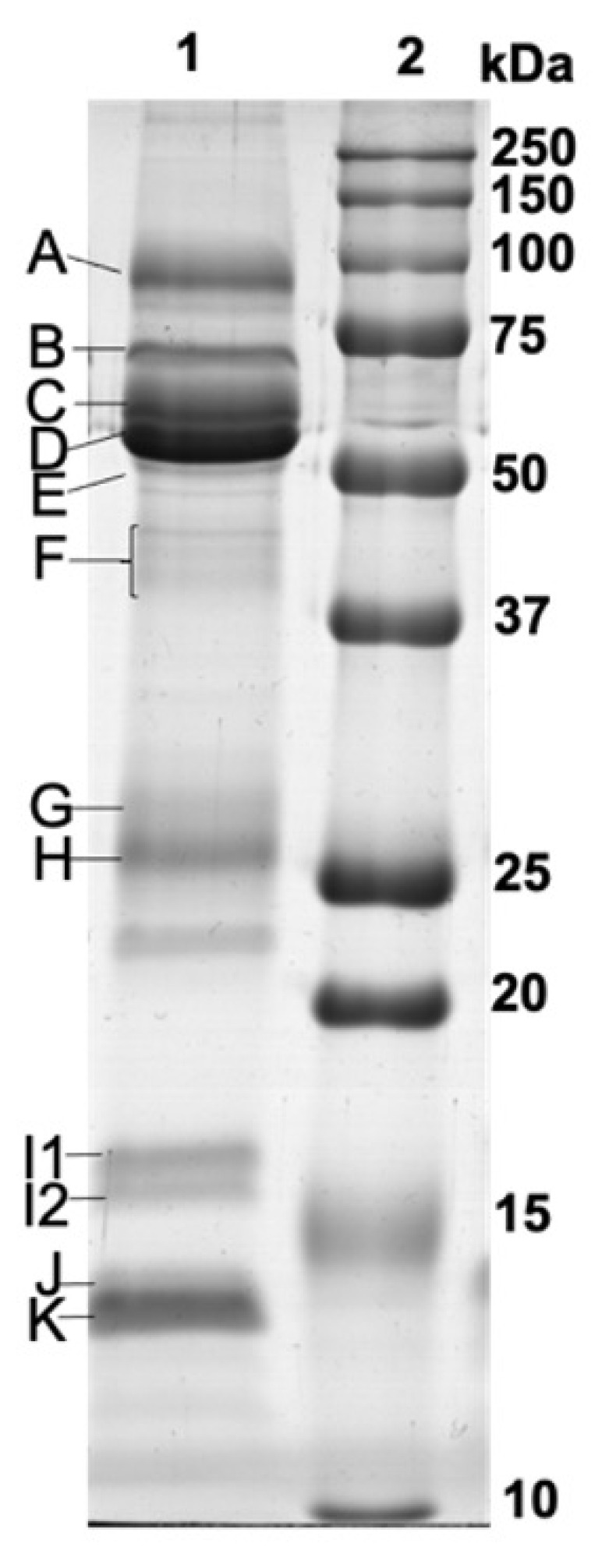
Representative salivary protein profile (1—saliva samples; 2—molecular mass marker; letters represent the 12 different protein bands analyzed).

**Figure 3 nutrients-13-01246-f003:**
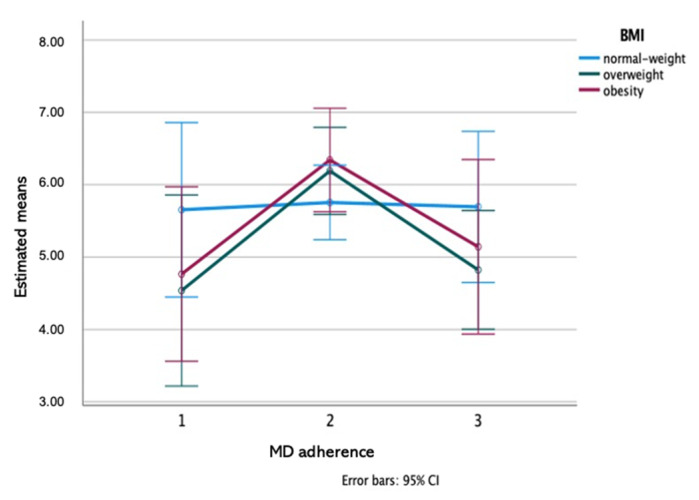
MD adherence*BMI interaction plot for salivary band K: a two-way ANOVA (*n* = 122; MD adherence effect *p* = 0.005; Eta partial squared = 0.09; power of the test = 85%).

**Figure 4 nutrients-13-01246-f004:**
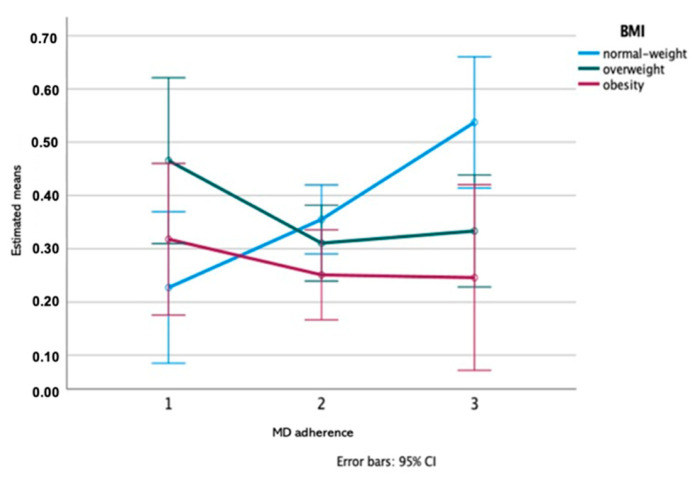
MD adherence*BMI interaction plot for salivary Amy_protein concentration: a two-way ANOVA (*n* = 122; MD adherence*BMI interaction effect *p* = 0.019; Eta partial squared = 0.11; power of the test = 80%).

**Figure 5 nutrients-13-01246-f005:**
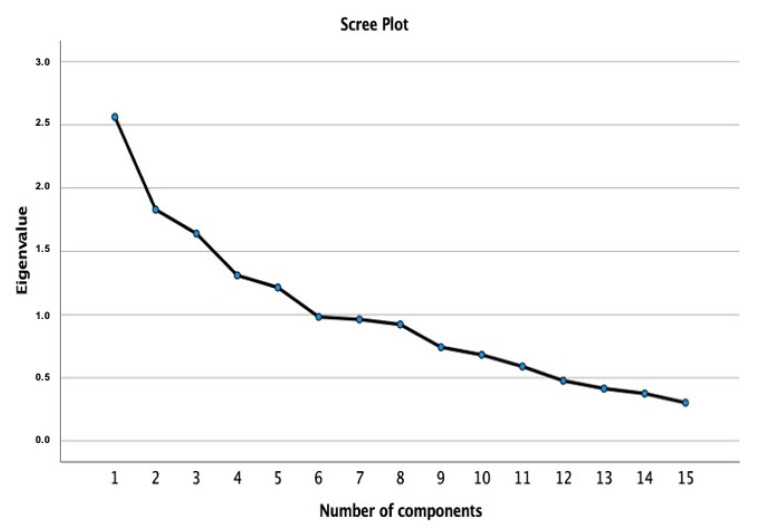
Scree plot used for visual inspection.

**Table 1 nutrients-13-01246-t001:** Variations in saliva composition between Mediterranean Diet (MD) adherence and Body Mass Index (BMI) groups (means ± SD): a two-way general linear model.

MD Adherence	*n*	Band A	Band B	Band C	Band D	Band E	Band F	Band G	Band H	Band I1	Band I2	Band J	Band K	Amy_ Protein (U/mg)
Group 1 (low)	18	5.7 ± 1.2	7.7 ± 2.3	9.8 ± 2.2	8.1 ± 1.7	2.1 ± 0.8	7.5 ± 1.3	4.4 ± 0.8	7.2 ± 1.5	3.4 ± 1.0	3.2 ± 1.3	7.5 ± 2.7	5.0 ± 1.1 ^A^	0.3 ± 0.2
Group 2 (medium)	76	5.7 ± 1.4	7.1 ± 2.2	9.0 ± 1.8	8.4 ± 2.7	1.8 ± 0.4	8.1 ± 1.3	4.3 ± 0.7	7.4 ± 1.5	3.5 ± 0.7	3.1 ± 0.8	8.1 ± 2.7	6.0 ± 1.5 ^B^	0.3 ± 0.2
Group 3 (high)	28	5.7 ± 1.6	7.0 ± 2.5	9.4 ± 2.4	8.6 ± 2.4	1.8 ± 0.4	7.8 ± 1.3	3.9 ± 0.9	7.4 ± 1.5	3.4 ± 0.7	3.2 ± 0.8	7.9 ± 1.9	5.1 ± 1.7 ^A^	0.4 ± 0.2
*p*-value														
MD adherence		0.880	0.597	0.506	0.947	0.206	0.389	0.093	0.883	0.931	0.600	0.615	**0.005 ***	0.324
BMI		0.649	0.816	0.924	0.146	0.284	0.921	0.203	0.213	0.145	0.471	0.675	0.431	0.108
MD adher × BMI		0.292	0.230	0.966	0.139	0.077	0.930	0.428	0.178	0.330	0.142	0.292	0.264	**0.019 ***

SD, standard deviation; BMI, body mass index. * *p* < 0.05; Two-way ANOVA; Bonferroni’s post-test.

**Table 2 nutrients-13-01246-t002:** Component loadings obtained by principal component analysis and Oblimin rotation of the saliva composition and flavonoids and polyphenol intake.

	Components				
	1	2	3	4	5
band A			0.768		
band B	−0.442				
band C	0.581		0.538	0.320	
band D	0.761				
band E					0.639
band F	0.449	−0.435		0.438	
band G	−0.307				0.478
band H			0.661		−0.367
band I1					0.628
band I2					0.635
band J				0.799	
band K	−0.344		−0.435	0.373	−0.338
flavanoids intake		0.754		0.403	
polyphenol intake		0.761			
Amy_protein (U/µg)	0.778				

Rotation converged in 15 iterations. Only coefficients greater than 0.30 are shown.

## Data Availability

The data presented in this study are available on request from the corresponding author. All data relevant to the study are included in the article and access to raw data would be provided upon request.

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
