# Peer review of "Relationship between Mediterranean Diet Adherence and Saliva Composition"

_nutrients, 2021, doi:10.3390/nu13041246_

Round 1
Reviewer 1 Report
Authors of the present paper aimed at assessing the variations in salivary protein profile related with individuals’ intake frequency of polyphenols.
The article presents quite interesting results, but despite this, there are some issues that authors should address.
In the Abstract, line 27, this is the aim and not methods
As the main purpose of the paper was to focus on flavanols authors should insert some description of flavanols.
The aim, as stated at the end of the Introduction, is unclear at least to me. The authors should clearly state their primary aim and specify if there is also a secondary aim.
Statistical analysis is difficult to follow if someone is not an expert. Then, even result section need to be well organized and described as it is very difficult to follow.
The Discussion section needs a better organization; authors first should briefly describe their results and then interpret them in light of what was already known.
It’s not clear, at least to me, why authors didn’t perform a taste test as in the Introduction they wrote: “Being some types of polyphenols, such as flavanols, particularly bitter and/or astringent, the richness in these compounds, although having health advantages, may limit sensory quality”.
As presented, the article cannot be published and it needs careful editing to improve the flow of information, correct word usage, etc.
Author Response
Authors of the present paper aimed at assessing the variations in salivary protein profile related with individuals’ intake frequency of polyphenols.
The article presents quite interesting results, but despite this, there are some issues that authors should address.
Authors - We would like to thank the reviewer for considering this results interesting. We corrected the manuscript and hope that the issued referred could be addressed.
In the Abstract, line 27, this is the aim and not methods
Authors - Thank you for the correction. We changed accordingly and included the methodology. Changes in some sentences were made to respect the limit of words of abstracts in Nutrients.
As the main purpose of the paper was to focus on flavanols authors should insert some description of flavanols.
Authors - Information was added to introduction.
The aim, as stated at the end of the Introduction, is unclear at least to me. The authors should clearly state their primary aim and specify if there is also a secondary aim.
Authors - Thank you for the repair. We agree that objectives were not clearly present. We re-write that part of the introduction, considering one general and two specific objectives.
Statistical analysis is difficult to follow if someone is not an expert. Then, even result section need to be well organized and described as it is very difficult to follow.
Authors - Statistical analysis and results section were corrected in order to clarify the type of analysis done and the results obtained from it.
The Discussion section needs a better organization; authors first should briefly describe their results and then interpret them in light of what was already known.
Authors - We thank the reviewer for the comment. Some adjustments were made to discussion section for better organization.
It’s not clear, at least to me, why authors didn’t perform a taste test as in the Introduction they wrote: “Being some types of polyphenols, such as flavanols, particularly bitter and/or astringent, the richness in these compounds, although having health advantages, may limit sensory quality”.
Authors - We understand the reviewer concern, and, in fact, we did collect data about the gustatory function of these individuals. However, due to the fact of relationship between gustatory function and dietary habits exist, but being not perfect (i.e., other factors affecting also intake besides taste sensitivity) and also to the fact of to exist a relationship between saliva composition and taste sensitivity, but again, not perfect (depending on taste, BMI, sex), we considered that it would add confusion to the article, instead of providing comprehensive information. As such, we opted by presenting only data about the relationship between salivary protein composition and dietary habits.
As presented, the article cannot be published and it needs careful editing to improve the flow of information, correct word usage, etc.
Authors - We answered the different points raised by the reviewer and tried to address them all. Moreover, English was corrected.
Reviewer 2 Report
I recommend introducing corrections, especially in the discussion of the results.

Author Response
I recommend introducing corrections, especially in the discussion of the results.
v. 54: Debatable. Drinking even small amounts of alcohol leads to cirrhosis of the liver. This disease is very common in nations where large amounts of wine are consumed. High resveratrol content has lower health-promoting effects than the presence of alcohol.
Authors - We agree with the reviewer, acknowledging that excessive consumption of alcohol is dangerous, particularly for the liver and cardiovascular health, and whether lower levels of alcohol consumptions offer any protection against coronary heart disease (CHD) remains controverse. Furthermore, although the French paradox has been attributed to the consumption of red wine and the intake of wine compounds such as resveratrol, this hypothesis was challenged and other explanations have been introduced, considering that the antioxidant activity of resveratrol may be negligible with respect to healthy benefits of red wine.
In the sentence, referred by reviewer, we only want to state that wine, in low to moderate levels, are present in MD, contributing also with polyphenols. We made some changes in the sentence to avoid this can be confounded with a statement of positive effects of wine consumption.
v. 66-73: The relationship of proteins contained in saliva with tannins has been studied for a long time and there are many scientific reports on this subject.
Authors - We agree. We corrected the sentence to highlight that.
v. 75-82: I don't understand why this particular group of polyphenols was chosen. When it comes to tartness and bitterness, tannins would be better. Flavanols are not predominantly found in the Mediterranean diet. Most often they form complex compounds in the form of tannins. It would also be interesting to study the resveratrol, which is present in grape skins and wine.
Authors - We thank the reviewer for the comment. We agree that choosing tannins would be interesting. However, for assessing polyphenol consumption we collected data through a questionnaire for 50 fruits and vegetables and converted the medium daily amount of that foods to polyphenol amounts, using the Phenol data explorer. Depending the type of food, this database has more or less detail in terms of individual phenolic components, so it was not easy to assess the amounts of tannins for all foods. For that reason, we opted by analyzing a group of polyphenols that contains bitter and astringency molecules. This, assuming that some of the salivary changes induced by polyphenols may be at the level of salivary proteins involved in bitterness and astringency acceptance. So, we wanted to test the hypothesis that, among the total polyphenols, could be those molecules, with such sensory characteristics, the ones with higher influence in saliva.
v. 86: The study group was quite small. What was the age distribution of the study participants in it? After all, it had a big impact.
Authors - Although we worked with 122 individuals, we understand that the reviewer considers a limited number of participants, since they are separated by 3 group of MD adherence, which are composed by a different number of participants, resulting in a N not high for some groups. We added this limitation to conclusions.
v. 271: This is not a new conclusion. It has already been investigated.
Nicola J. Baxter, Terence H. Lilley, Edwin Haslam, and Michael P. Williamson, 1997: Multiple Interactions between Polyphenols and a Salivary Proline-Rich Protein Repeat Result in Complexation and Precipitation, Biochemistry, 36, 18, 5566–5577
Anders Bennick, 2002: Interaction of Plant Polyphenols with Salivary Proteins, Critical Reviews in Oral Biology & Medicine, 13(2):184-196
Susana Soares, Rui Vitorino, Hugo Osório§, Ana Fernandes, Armando Venâncio ∥ , Nuno Mateus, Francisco Amado, and Victor de Freitas, 2011: Reactivity of Human Salivary Proteins Families Toward Food Polyphenols, J. Agric. Food Chem. 2011, 59, 10, 5535–5547Author - We are not sure if we could identify the part of discussion reviewer is referring to, since in the pdf created during submission process, line 271 corresponds to one of the lines from Table 2. Taking into account the comment, we believe that the reviewer is referring to line 279, where we present our hypothesis about variations in salivary protein profile according to diet composition.
The articles the reviewer present are all related with the existence of a relationship between salivary proteins and dietary polyphenols, but in the context of an interaction between these molecules when both are present. However, none of these studies analyzed if the levels of the salivary proteins able to interact with tannins (or others) are different according with the levels of these compounds that make part of the diet. In animals it has been shown that the presence of tannins results in changes in saliva composition (and in that case the changes are pointed as a defense mechanism). In humans, from our knowledge, this has been few experimented.
We had this information, in that part of discussion, to address reviewers’ concern.
v.302-306: Why is the discussion about tannins? In the work, the authors should focus on flavanols.
Authors - In this part the authors did try to explain the observed association between salivary cystatin levels and wine intake, that was the item (from the MEDAS score) that presented a significant correlation with the levels of these salivary proteins. We hypothesized that this relationship could be due to the particular high levels of tannins that red wine contains, since a relationship between the levels of these salivary proteins and tannins has been previously reported in different studies. This discussion was directed to the association salivary cystatins*wine and not for the association between salivary cystatins*flavanols that was observed in another analysis.
We added one sentence in this part of discussion, to clarify it.
v. 351-354: These are too far-reaching conclusions. No other factors such as age were taken into account.
Authors - We agree with the reviewer and corrected the sentence to direct to MD pattern.
Reviewer 3 Report
The present paper aims to evaluate to what extent the adherence to a Mediterranean dietary pattern is associated with polyphenol intake and with saliva composition in humans. All the manuscript sections are well discussed and show interesting data, although novelty from the experimental point of view was not noticed. In addition, manuscript should be checked by a native speaker since English quality is not the best.
Other comments:
Results
- The Authors should improve the quality of Figure 1, it looks too grainy
- Table 1 could be more understandable if you add the ± symbol before each SD value
Discussion
- Line 283: "Associated" instead of "Association"
Author Response
The present paper aims to evaluate to what extent the adherence to a Mediterranean dietary pattern is associated with polyphenol intake and with saliva composition in humans. All the manuscript sections are well discussed and show interesting data, although novelty from the experimental point of view was not noticed.
Authors - We would like the reviewer for finding our results interesting and well discussed. Although saliva has been considerable investigated in the context of polyphenol intake, in humans this has been less done than in animals and, from our knowledge not previously related with MD adherence.
In addition, manuscript should be checked by a native speaker since English quality is not the best.
Authors - English has been reviewed
Other comments:
Results
- The Authors should improve the quality of Figure 1, it looks too grainy
Authors - An image with higher quality was introduced in the document.
- Table 1 could be more understandable if you add the ± symbol before each SD value
Authors - Done
Discussion
- Line 283: "Associated" instead of "Association"
Authors - We think it was in line 342 – Correction done
Round 2
Reviewer 2 Report
I recommend the article for publication.